# Lamellar carbon nitride membrane for enhanced ion sieving and water desalination

Yang Wang [1] ✉, Tingting Lian [1], Nadezda V. Tarakina [1], Jiayin Yuan [2] & Markus Antonietti [1] ✉

Membrane-based water treatment processes offer possibility to alleviate the water scarcity dilemma in energy-efficient and sustainable ways, this has been exemplified in filtration membranes assembled from two-dimensional (2D) materials for water desalination purposes. Most representatives however tend to swell or disintegrate in a hydrated state, making precise ionic or molecular sieving a tough challenge. Here we report that the chemically robust 2D carbon nitride can be activated using aluminum polycations as pillars to modulate the interlayer spacing of the conjugated framework, the noncovalent interaction concomitantly affords a well-interlinked lamellar structure, to be carefully distinguished from random stacking patterns in conventional carbon nitride membranes. The conformally packed membrane is characterized by adaptive subnanochannel and structure integrity to allow excellent swelling resistance, and breaks permeability-selectivity trade-off limit in forward osmosis due to progressively regulated transport passage, achieving high salt rejection (>99.5%) and water flux (6 L m$^{-2}$ h$^{-1}$), along with tunable permeation behavior that enables water gating in acidic and alkaline environments. These findings position carbon nitride a rising building block to functionally expand the 2D membrane library for applications in water desalination and purification scenarios.

Two-dimensional (2D) materials stacked by atomic-thick layers hold promise for permselective membrane assemblies to tackle water scarcity and contamination[1–3]. The interlayer channels or intralayer pores of 2D membranes in the subnanometer range constitute the dominant transport pathways for precise molecular and ionic sieving[4,5], along with intriguing nanofluidic phenomena that are otherwise absent in their bulk counterparts[6]. While representative 2D materials (e.g., graphene oxide, MXene) have been promisingly explored in water desalination, molecular filtration, and purification[7,8], their rich oxygenated hydrophilic groups dangling on nanosheet edges or in between adjacent layers render serious swelling in an uncontrollable manner or even disintegration when these membranes are exposed to aqueous or organic solutions/vapors[9,10]. This is also related to the fact that in subnanometer channels, swelling forces and

normal forces due to flow can become enormous[11], and only the most rigid, well-interlinked structures are expected to handle this swelling stress for subnanometer sieving that requires high accuracy.

Tracing back to the large 2D materials family, polymeric carbon nitrides (CNs, also widely termed as g-C$_3$N$_4$) are predominantly applied as semiconductors to catalyze photochemical reactions[12,13]. In terms of membrane assembly, the chemically and thermally stable CNs, when exfoliated into flat sheets, would be ideal layer components to withstand interlayer swelling in liquid environments. The emerging progress has also propelled strategies to be developed for CN membrane fabrication (either in free standing or substrate-coated form) and its use in photoelectrochemical and ion diffusion cells, including microwave-assisted condensation, microcontact printing, chemical vapor deposition, liquid-based and direct growth, etc[14–19]. In this way,

[1]Department of Colloid Chemistry, Max Planck Institute of Colloids and Interfaces, 14476 Potsdam, Germany. [2]Department of Materials and Environmental Chemistry, Stockholm University, 10691 Stockholm, Sweden. ✉e-mail: Yang.Wang@mpikg.mpg.de; Markus.Antonietti@mpikg.mpg.de

even more relevant to wider applications than cell performance improvement is the inhomogeneous microstructure and nanostructure governed by the interplay between CN and substrate. Inspired by the liquid exfoliation of typical layered materials, CN membrane assembly via filtration-assisted coating of as-exfoliated nanosheets onto porous filter is seemingly plausible, which has been tentatively explored for filtration and separation purposes with improved homogeneity at microscale[20-22]. However, recognized efforts only include the removal of large-sized dyes, organic pollutants, and nanoparticles using CN membranes in ultrafiltration and nanofiltration processes[23,24], occasionally coupled with photocatalytic reactions[25,26]. In reverse osmosis and forward osmosis, CN nanosheets only serve as a modifier or filler that incorporates into thin-film nanocomposite and thin-film composite membranes[27-31]. It is then noted that the random alignment of CNs with its rather interactive tri-s-triazine-based motifs as building subunits, can hinder foreign intercalants to access and tune the interlayers. In this context, subnanochannel activation with elaborate control to provide transport passage for molecular and ionic species of interest remains unattainable for CNs to be used in water desalination membranes with precise sieving ability.

Pristine CNs-texturally similar to graphene and its derivatives-are also characterized by ~3.2 Å stacking distance or gallery height along the c-axis[32-34], which falls in the range between the size of water (diameter: 2.8 Å) and hydrated salt ions (exemplary diameter: 6.62 Å for K$^+$)[35]. The

CN membranes are theoretically qualified for water transport and to block unwanted salt species at the subnanometer scale, but experimental results were counterintuitive and evidenced that their permeation behavior substantially deviates from such a perfect transport passage[20]. Prevailing explanations are structural defects that create excess pathways for larger molecules or ions to access[21,23], but microstructural changes throughout the membrane assembly process have long been ignored. The exfoliated CNs are indeed rather stiff and flat 2D sheets[36], but their traditional membrane microstructure reflects more of a random restacking behavior, which is not in line with an unperturbed 2D lamellar tectonic unit[22,24]. The CN nanoflakes are found seriously wrapped or wrinkled under external pressure (i.e., vacuum filtration) and fail to realize conformal packing in a tight 2D fashion (Fig. 1a).

In the present contribution, we propose an approach that could activate and stabilize the transport channels of CNs at the subnanometer level whilst constituting a lamellar membrane structure (Fig. 1b). The rigid and well-interlocked nanostructure, arising from the strong but tunable noncovalent interaction between foreign pillaring agent and CN, is presented with adaptive subnanochannel of <6 Å in width. The conformally packed membrane is highly stable in water and shows excellent anti-swelling properties, with minimal fluctuation of interlayer spacing in dry and hydrated states (5.6–5.9 Å). Unlike the disordered packing mode of conventional CN membrane with ubiquitous structure defects, the regular transport passage of our 2D

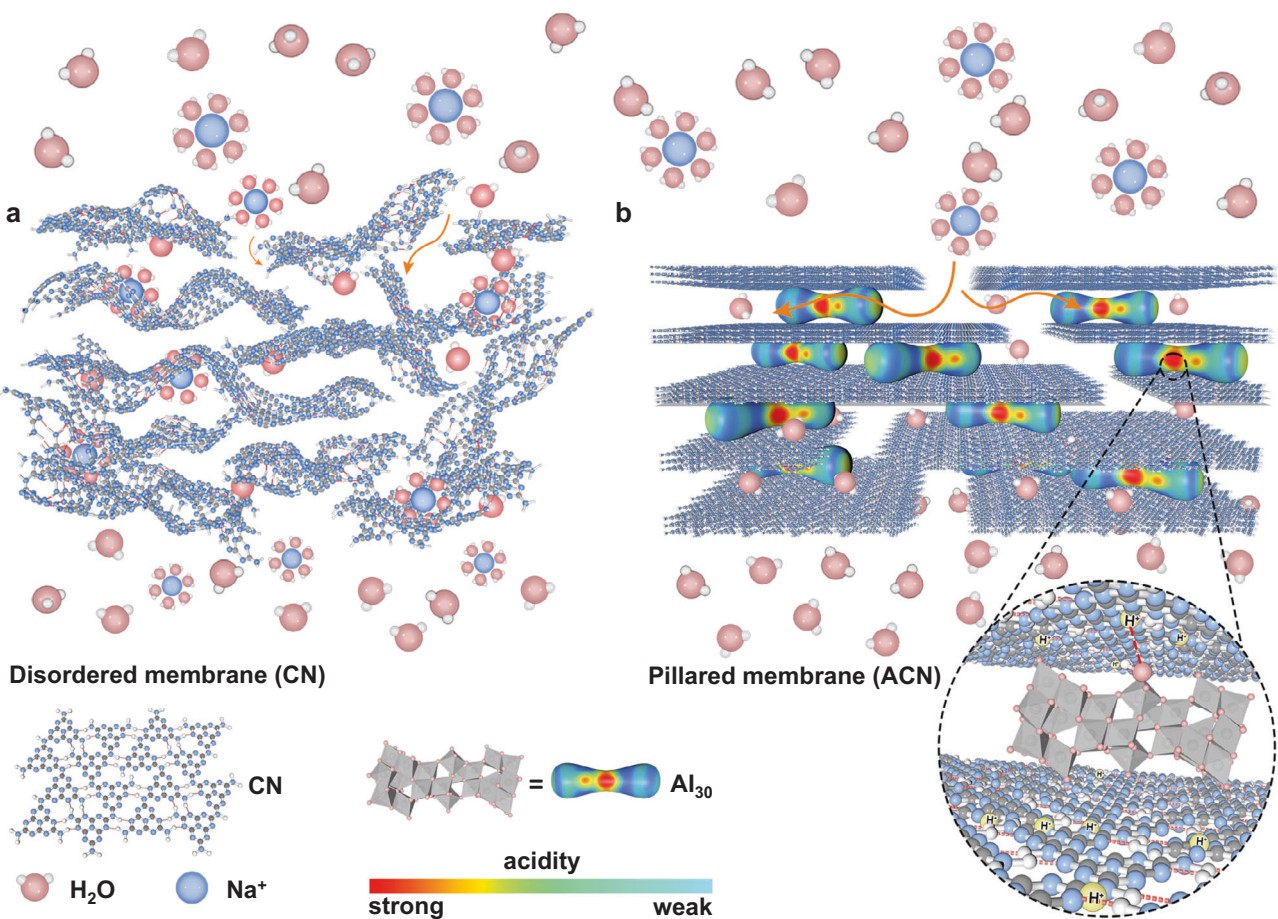

**Fig. 1 | Schematic of two membrane configurations for molecular and ionic permeation. a** Disordered membrane (CN) is represented by random stacking behavior, which allows the concomitant transport of water and ions. **b** Intercalated by the Al$_{30}$ polycations with asymmetric distribution of acidic sites, the pillared membrane (ACN) with conformal packing mode shows lamellar structure,

accompanied by the formation of expanded interlayer spacing, which allows water transport but blocks salt ions. The interaction between Al$_{30}$ and CN is highlighted in the round frame, nitrogen sites in CN are protonated and share hydrogen bonding with Al$_{30}$ to interlock the layers.

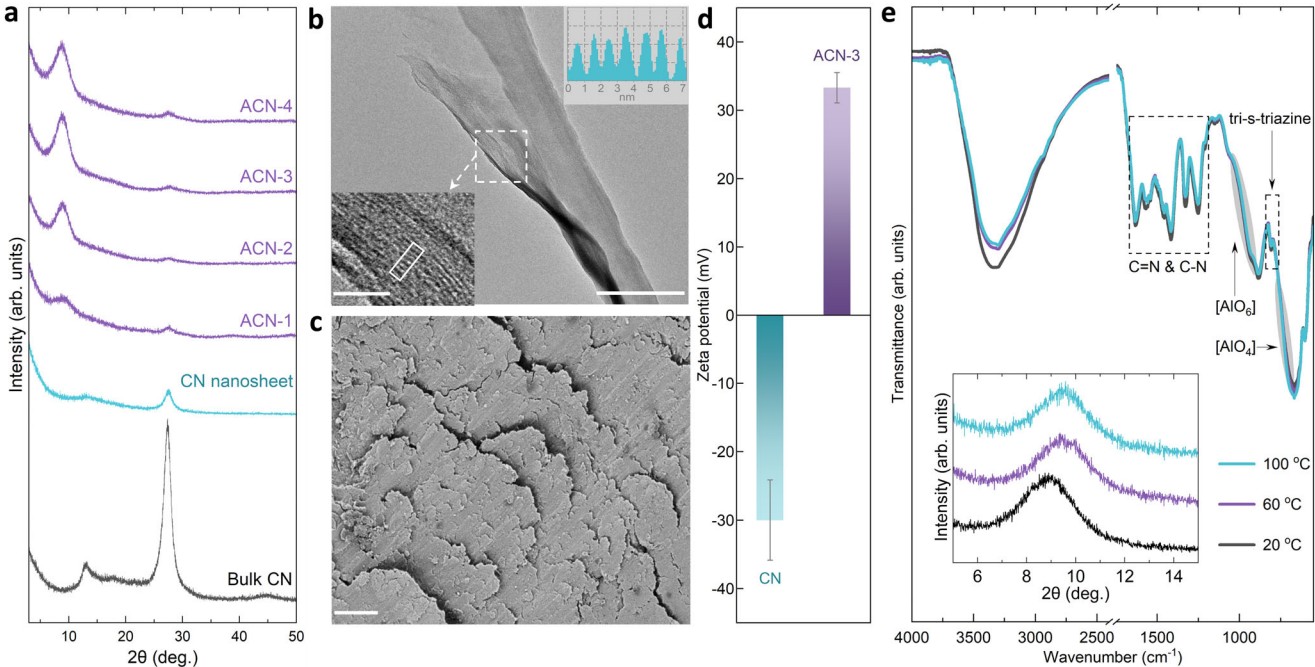

**Fig. 2 | Structure evolution. a** XRD patterns of pristine bulk CN, exfoliated CN nanosheets and ACN samples. ACN-1, 2, 3, 4 represent the initial mass ratios of $Al_{30}$ to CN, which are 0.25:1, 1:1, 2:1, and 4:1, respectively. **b** TEM image of ACN-3 powder, scale bar: 100 nm. Inset: high-resolution TEM image of the marked square in **b** (bottom panel, scale bar: 10 nm) and TEM contrast intensity (top panel) of the rectangle region in bottom panel. **c** Scanning electron microscope (SEM) image of ACN-3 powder, scale bar: 5 μm. **d** Zeta potentials of CN nanosheet and ACN-3 dispersions. **e** FT-IR spectra (inset: XRD patterns) of ACN-3 powder dried at elevated temperature under vacuum, the vibrations assigned to CN and $Al_{30}$ are marked by dashed rectangles and gray ellipses, respectively. Error bars in **d** represent the standard deviations of three independent measurements.

lamellar CN membrane allows high water flux (6 L m⁻² h⁻¹), which concomitantly occurred with high salt rejection (99.5%) in forward osmosis, breaking permeability-selectivity trade-off limit and outperforming the widely reported 2D membranes. The membrane maintains structural integrity in both acidic and alkaline environments, upon which smart water gating performance can be achieved with minimal loss of salt rejection rate.

## Results

### Subnanochannel activation and conformal packing of CNs

The tri-*s*-triazine building units are primary motifs interconnected by secondary nitrogen to constitute the conjugated 2D plane of CN, which is further bridged by in-planar hydrogen bonds[32,33] (Fig. 1). Previous endeavors using concentrated acid or base for top-down cleavage of CNs may facilitate intercalation and render highly dispersed colloidal suspensions for membrane preparation, but the accompanying partial chemolysis produces pore defects or one-dimensional derivatives (quantum dots, nanorods, nanofibers, etc.), thus hindering a wider deployment at subnanoscale[22,29,37]. The long-pair electrons on nitrogen atoms of the framework, along with the -NH groups that bridge adjacent motifs and −NH₂ groups at edge positions, endow CNs with abundant Lewis basic sites for further functionalization (Fig. 1b) at mild conditions. We apply a Keggin cluster polycation $[Al_{30}O_8(OH)_{56}(H_2O)_{24}]^{18+}$ (abbreviated as $Al_{30}$) as pillaring agent[38] to interact with the exposed electron-rich, basic sites and intercalate in between adjacent CN layers. The high-charge $Al_{30}$, characterized by the asymmetric distribution of acidic sites[39] and highest acidity centered at the equatorial region (Fig. 1, Supplementary Fig. 1, sites 3 and 4 of bound water, η-H₂O), is found to interact with CN at much higher adsorption energy than the lower acidic $[Al_{13}O_4(OH)_{24}(H_2O)_{12}]^{7+}$ cluster (abbreviated as $Al_{13}$) despite its smaller size (9 × 9 × 9 Å)[40] and lower steric hinderance (Supplementary Figs. 1–11, Supplementary Note 1).

Pristine bulk CN features a characteristic graphitic stacking peak at 27.5° (002), while the second peak at 13.3° (100) is assigned to in-

plane repeating units (Fig. 2a). Exfoliated CN nanosheets and pillared $Al_{30}$-CN composites (abbreviated as ACN) show a dramatically weakened graphitic peak intensity[41], which quantifies an efficient exfoliation process following established procedures. A new peak centered at ~9° is identified and intensified by steadily increasing $Al_{30}$ amount, coupled with further attenuation of the pristine stacking peak. This implies that the pristine interlayer distance (~3.2 Å) of CN is substantially widened to 9.8 Å upon $Al_{30}$ intercalation. Consistently, high-resolution transmission electron microscope (HR-TEM) images (Fig. 2b) reveal the thin-layer morphology of ACN with resolvable fringe separation, which corresponds to an interlayer spacing similar to that of X-ray diffraction (XRD) result, the ordered stacking mode is however not observed in conventional CN nanosheets and $Al_{13}$-CN ones (Supplementary Fig. 12). Given the thickness of ideal single-layered CN (3.2 Å), the free spacing of ACN for molecules and ions to access is calculated as 6.6 Å, close to the size of hydrated monovalent salt ions. $Al_{30}$ with nominal geometric dimensions of 10 × 10 × 20 Å[40], even lying flat in between CN layers with optimal configurations (Supplementary Fig. 5), is assumed to afford larger free spacing of ~10 Å, but the loss of water molecules during the post-drying process can explain the smaller-than-expected value. The restacked ACN nanosheets in a powder form already show a lamellar texture (Fig. 2c), which is easily distinguished from the restacked CN nanosheets with random particle-like morphology (Supplementary Fig. 9). The negative charge of CN nanosheets is neutralized and the ACN-3 composite then carries net positive charge at higher loading amount of the polycationic pillar $Al_{30}$ (Fig. 2d), which allows electrostatic interaction and hydrogen bonding with CN (Supplementary Figs. 13–14, Supplementary Table 1, Supplementary Note 2). Fourier-transformed infrared (FT-IR) spectra confirm the coexistence of CN and $Al_{30}$ in ACN composite, with progressively decreased intensity of O-H band vibration (3200–3500 cm⁻¹) at higher temperature (Fig. 2e). The free spacing, upon partial removal of water molecules[42,43], is narrowed from 6.6 Å (20 °C) down to 6.1 Å (60 °C) and then stabilizes even at 100 °C (inset

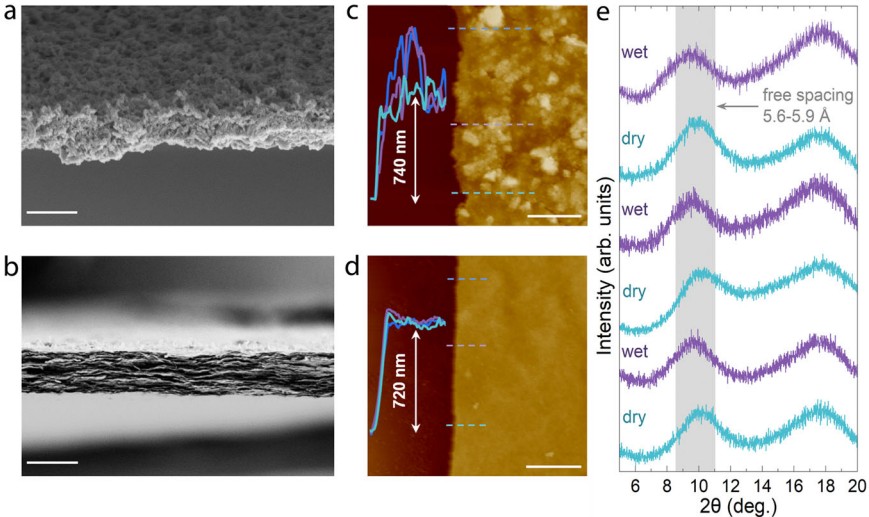

**Fig. 3 | Membrane morphology and stability. a, b** Cross-sectional SEM images of **a** CN and **b** ACN-3 membrane, scale bar: 1 μm. **c, d** AFM images of **c** CN (scale bar: 10 μm) and **d** ACN membranes (scale bar: 20 μm) with corresponding height profiles (inset). **e** XRD patterns of ACN-3 membranes subjected to drying-wetting cycles,

each drying or wetting event lasts for 24 h. The gray region marked in **e** represents the free spacing fluctuating within 5.6–5.9 Å in dry or wet state, the broad diffraction peak centered within 2θ = 16–20° is assigned to the polydopamine-coated polyether sulfone filter.

of Fig. 2e). The combined findings collaborate the pillaring effect of Al_{30}, which activates transport channel at subnanometer scale and concurrently promotes the conformal packing of CN nanosheets.

### Assembly and stability assessment of lamellar ACN membrane

It is then coming with the membrane assembly by vacuum filtration of the freshly prepared ACN suspension and subsequent in situ washing to remove excess Al species (Supplementary Fig. 15). The membrane thickness can be controlled by changing the volume of the filtered suspension (Supplementary Fig. 16). Cross-sectional scanning electron microscope (SEM) images show typical layer-by-layer stacking pattern of ACN membrane, while a random particulate-like morphology is found for the pristine CN membrane (Fig. 3a–b, Supplementary Fig. 17), a common result in previous reports which was used to explain the transportation of molecules and ions based on a hypothesized 2D structure[21,30,31]. The heterogeneity reflected by thick- and thin-coated regions can potentially generate cracks when CN membranes are scaled up. Enabled by polycation pillaring to provide conformal packing, the lamellar ACN membrane comes with a flat surface relative to that of CN, the latter fluctuating substantially with high roughness (Fig. 3c–d). The heterogenous and homogenous coating behavior also holds true for thicker CN and ACN membranes (~1500 nm), respectively (Supplementary Fig. 18). Considering the high proportion of hydroxy/aqua ligands of Al_{30} retained in ACN after drying at 20 °C, we then baked these membranes at elevated temperatures (up to 100 °C) to stabilize the nanostructure of ACN membrane. Similar to the powder form, the free spacing of dried membranes contracts slightly (6.3 Å at 20 °C, 5.6 Å at 60 °C, and 5.3 Å at 100 °C), i.e., the bound water was an integral part of the membrane structure. The conformally packed membrane baked at 60 °C shows excellent swelling resistance when immersed in water and aqueous salt solutions, as evidenced by the minimal change of free spacing (Fig. 3e, Supplementary Figs. 19–25, Supplementary Note 3) in wet or dry state, which fluctuates within a narrow range of 5.6–5.9 Å. We also observed that 60°-drying enables the efficient removal of weakly-bonded hydrophilic groups and switches the membrane surface from hydrophilic to hydrophobic (Supplementary Fig. 20), leaving some bound water molecules confined in the subnanochannel. The appreciable stability of ACN membrane is also revealed by multiple drying-wetting cycles of the membrane (Fig. 3e). The exposed groups in ACN, although hydrophilic, are firmly coordinated with the inorganic cluster surfaces. This

explains the rather rigid structure of the subnanochannels in the final membranes, where the Al-crosslinking of CN layers finally locks into a cooperative and homogeneous 2D morphology, as confirmed by the argon-ion-sputtered X-ray photoelectron spectra with etching depth up to 200 nm (Supplementary Fig. 24).

### Ion sieving and water desalination of CN and ACN membranes

Next, we evaluated the ion permeation behavior through the assembled membranes (supported by a polydopamine-coated polyether sulfone filter to enhance the adhesion with membranes) in a customized H-shaped cell equipped with two reservoirs (Supplementary Fig. 26a). This was firstly exemplified using NaCl as feed solution. The pristine CN membrane (thickness: 740 nm) shows a high permeation rate of Na^+ up to 1.64 mol m^{-2} h^{-1} (Fig. 4a), which barely follows a thickness-dependent manner and only decreases to 1.39 mol m^{-2} h^{-1} when the membrane thickness is doubled (Supplementary Fig. 27). This further verifies the defecteous nature of at least of our own reference membrane, but we assume this to be general. With increasing Al_{30} amount, the regular subnanochannels (~5.9 Å in wet state) are gradually activated for ACN membranes with the similar thickness (720 nm), leading to progressively lowered Na^+ (hydrated diameter: 7.2 Å) permeation rate, which decreases to 3.14 × 10^{-3} mol m^{-2} h^{-1} in ACN-3 and stabilizes for ACN-4 (Fig. 4a). Other monovalent and divalent species (K^+, Li^+, Ca^{2+}, and Mg^{2+} with hydrated diameters of 6.6, 7.6, 8.2, and 8.6 Å, respectively) in CN membranes also possess high permeation rates of 1.39–1.93 mol m^{-2} h^{-1} (Fig. 4b), underlying the non-selective barrier inside the randomly stacked nanostructure and coupled transport behavior dominated by larger defects. This contradicts also the widely claimed subnanochannel in pristine CN, which up to now only blocked large-sized dye molecules or nanoparticles in experimental outputs[23,24]. Similarly, the Al_{13}-CN membranes (thickness: 750 nm) also show comparable permeation rates of these ions with that of CN ones (Supplementary Fig. 28), due to the absence of activated transport channels.

In comparison, ACN-3 membrane quantifies substantially decreased permeation rates at levels of 10^{-3} mol m^{-2} h^{-1} for K^+, Li^+ and of 10^{-4} mol m^{-2} h^{-1} for Ca^{2+} and Mg^{2+} (Fig. 4b). That is, the ion permeation behavior is governed by the size-exclusion effect when the activated subnanochannel shows narrower free spacing than the hydrated diameters of these cations (Supplementary Table 2). The lower permeation rate of divalent species relative to the monovalent

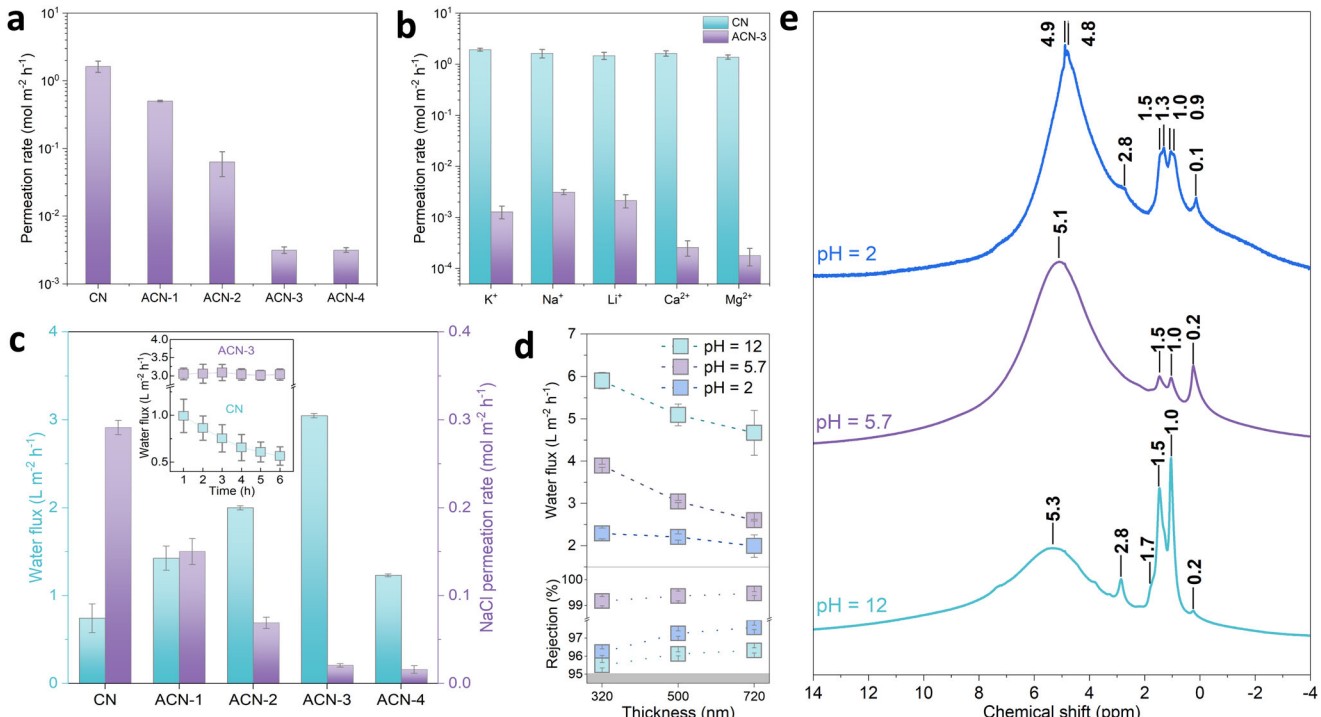

**Fig. 4 | Ion permeation behavior and water desalination performance of CN and ACN membranes. a** Permeation rate of $Na^+$ in CN and ACN membranes. **b** Permeation rates of monovalent ($K^+$, $Na^+$, $Li^+$) and divalent ($Ca^{2+}$, $Mg^{2+}$) salt ions in CN and ACN-3 membranes. **c** Water flux and NaCl permeation rate in CN and ACN membranes in forward osmosis, inset: variation of water flux as a function of permeation time. **d** pH-dependent water flux and salt rejection of ACN-3 membranes with different thickness. **e** $^1H$ CP/MAS ssNMR spectra of ACN-3 sample after treatments under different conditions, the signals are roughly grouped into four subsets[50,67]: (1) Terminal OH groups (0.1–0.2 ppm), arising from the polymeric nature of $Al_{30}$ which contains monomers and oligomers. (2) Bridged $\mu_2$-OH groups (Al-OH-Al) with resonances in 1–3 ppm. (3) Bridged $\mu_3$-OH groups ((Al)$_3$-OH) with resonances in 3–4 ppm. (4) Bound (or adsorbed) water molecules and hydrogen-bonded OH groups in lower field (4–6 ppm). Error bars in **a**–**d** represent the standard deviations of three independent measurements.

ones is ascribed to the higher dehydration energy barrier[44,45]. This is further supported by the temperature-dependent ion permeation behavior (Supplementary Fig. 29), which follows the Arrhenius equation, $\exp(-E/k_BT)$, where $E$ is the energy barrier and $k_B$ is the Boltzmann constant[45]. The $E$ values of divalent ions (41.8 kJ mol$^{-1}$ for $Ca^{2+}$ and 44.8 kJ mol$^{-1}$ for $Mg^{2+}$) are much higher than that of monovalent ones (11.5, 13.7, and 13.1 kJ mol$^{-1}$ for $K^+$, $Na^+$, and $Li^+$, respectively). As such, the divalent species are less likely to enter into the subnanometer-scale nanochannels and therefore come with exponentially lower permeation rates. The ion rejection performance of ACN membrane was then examined with a step closer to the real-world scenario, where we used synthetic seawater with mixed ions as feed solution and also found a decrease of their permeation rates by at least 100 times relative to that of pristine CN membrane (Supplementary Fig. 30), in the highly saline environment. The stability of our ACN membrane is then analyzed by the linear variation of $Na^+$ permeation rate as a function of initial salt concentrations up to 2 M, which readily returns to the initial state upon abruptly decreasing the salt concentration in the feed compartment (Supplementary Fig. 31).

Water flux and salt rejection performance were next evaluated in a similar cell configuration with the feed and draw reservoirs filled with 0.1 M NaCl and 2 M sucrose to model a forward osmosis mode (Supplementary Fig. 26b), an energy-efficient process for water desalination. The average water flux of pristine CN membrane was 0.74 L m$^{-2}$ h$^{-1}$ (Fig. 4c), comparable to that of graphene oxide but lower than that of the commercial benchmark (5–10 L m$^{-2}$ h$^{-1}$)[45,46]. We found that water permeates through 500 nm-thick CN membrane with a gradually decreased rate within the permeation period, due to the presence of a minor amount of pores and defect structure of CN, which allows concomitant transport of NaCl and sucrose and lowers the osmotic pressure difference as driving force (inset of Fig. 4c,

Supplementary Fig. 32). This is emphasized, as previous CN-based membranes were highlighted for their high water flux[23,24,29], which, however, were not cross-tested for semipermeability considerations. The time-dependent water flux, arising from either materials heterogeneity or non-classical fluid transport, is only alleviated in the thick membrane (1500 nm) with high-tortuosity transport pathways (Supplementary Figs. 33–34).

Comparatively, our ACN membranes are capable of facilitating water transport, as evidenced by the steady water flux with prolonged permeation time (inset of Fig. 4c, Supplementary Figs. 35–36). Note that both CN and ACN are structurally connected by intraplanar hydrogen bonds to bridge the tri-s-triazine motifs (Supplementary Fig. 5); they are less likely to form intralayer pores that can allow water molecules to fit in properly. These "pores" can only be formed by rigid covalent C–N bonding of ideally stacked graphitic CN (g-C$_3$N$_4$), which have not been experimentally available to date[33]. Although it is seemingly possible to apply poly(triazine) imide (PTI) with regular triangular intralayer pores, the synthesized PTI is only presented with small crystallites rather than nanosheets, largely restricting their membrane-formation ability for separation purposes[47–49]. In this regard, the evidently enhanced water permeation of ACN membrane can be primarily ascribed to the regular subnanochannel activated by $Al_{30}$. Increasing the amount of $Al_{30}$ pillar in ACN membranes contributes to elevated water flux whilst suppressing NaCl permeation, breaking the permeability-selectivity trade-off (Fig. 4c). At optimized conditions, the 500 nm-thick ACN-3 membrane holds both high water flux of 3.0 L m$^{-2}$ h$^{-1}$ and NaCl rejection rate of 99.4% (Fig. 4d), which can be well maintained after alternate immersion treatments (Supplementary Fig. 37). The water flux can be further improved using thinner PES filter to alleviate internal concentration gradient, without sacrificing the salt rejection performance (Supplementary Fig. 38). Upon prolonging the

permeation time up to 72 h, we found that the ACN-3 membrane only experiences a slight drop of water flux and salt rejection, suggesting the favorable long-term feasibility that can potentially allow its use in forward osmosis (Supplementary Fig. 39).

## Tunable permeation behavior in ACN membrane

We then adjusted the pH values of the aqueous NaCl solution at the feed side and found that the membrane is more permeable for water in an alkaline (pH = 12) than in an acidic (pH = 2) environment, with a water gating ratio of 2.6 at 320 nm which decreases to 2.3 at 500 and 720 nm (Fig. 4d). This is accompanied by the slightly higher permeation rate of NaCl in both cases (pH = 2 and 12). We used $^1$H cross-polarization/magic angle spinning solid-state nuclear magnetic resonance (CP/MAS-ssNMR) spectra to probe the microenvironmental change of ACN nanostructure upon pH adjustment and to explain the altered permeation behavior (Fig. 4e). In pH = 2 sample, the high-field (4.8 ppm: hydrogen-bonded OH groups; 4.9 ppm: physically adsorbed water) signals[50–52] are of much narrower line width relative to the other samples (not deconvoluted here), indicating strong hydrogen bonds and trapped water[53,54] in between adjacent CN layers in an acidic environment. The specific contribution from the acid proton affords additional hydrogen bonding with the bridged OH groups and makes the resonance at 4.8 ppm resolvable. Although the subnanochannel in our ACN membrane (5.9 Å in the hydrated state) can accommodate water molecules with smaller size (2.8 Å), the as-formed hydrogen bonds block the contiguous pathway for the slip flow of water with high velocity. Besides, the higher proportion of bound water immobilized by the hydrogen bonds may also occlude the transport channel of ACN. Increasing pH results in downfield shifting from 4.9 ppm to 5.1 ppm (pH = 5.7) and 5.3 ppm (pH = 12), along with the decreased signal intensity that is more evident in pH = 12 sample. The resonance shift indicates that the trapped water gradually evolves into a mobile and/or free state[54], accompanied by the broadening of line width arising from the chemical exchange. Note that the single broad line in the two cases (pH = 5.7 and 12) does not exclude the absence of hydrogen bonds, which may overlap with the lower-field resonance of more active water, which, however, proceeds with a far less pronounced effect in neutral, and especially, in alkaline conditions, as evidenced from the substantially attenuated intensity. In this way, water transports through the subnanochannel with the lower barrier in a neutral environment and turns to be much less tethered in an alkaline medium, contributing to enhanced water flux.

Situated in an alkaline environment (pH = 12), the hydration shell of sodium ions is downsized by the charge shielding effect[55], which allows the partially dehydrated species to partition into and permeate through ACN layers. Deprotonation of the acidic $\eta$-H$_2$O in Al$_{30}$ at pH = 12 leads to a less negatively charged ACN (Supplementary Fig. 40) that can interact with Na$^+$ via electrostatic attraction. They collectively give rise to an increased NaCl permeation rate in our ACN membrane. The deprotonation process is accompanied by partial hydrolysis, as suggested by the new resonances relating to the OH groups (2.8 ppm and 1.7 ppm) on non-framework Al species[54,56–58], with higher intensity relative to that of pH = 2 sample. Although the hydration shell of Na$^+$ in neutral and acidic conditions are similar[55], we notice again a slightly increased NaCl permeation rate for ACN membrane in an acidic environment. In this case, Al$_{30}$ can serve as an alkali, and the bridged OH groups accept protons from acid; the acid-base neutralization causes the cleavage of hydroxyl bond, upon which the acid-catalyzed partial decomposition of Al$_{30}$ appears to be possible[59]. This is reflected by the newly emerged resonances assigned to non-bonded water monomers (1.3 ppm and 0.9 ppm)[54,60], stemming from the catalytically decomposed Al$_{30}$ paired with the formation of water[59]. The nominal positive charge of Al$_{30}$ (+18) is then decreased and the electrostatic repulsion between ACN and Na$^+$ is alleviated, giving rise to a slightly decreased salt rejection rate. (Supplementary Figs. 40–42, Supplementary Note 4). These variations,

however, come with only minor contributions, as evidenced by the high rejection rate of NaCl maintaining above 95% (Fig. 4d). Long-term operation of the ACN-3 membrane in acidic and alkaline environments for 72 h only results in a decrease of water flux within 10% and salt rejection within 2.8% (Supplementary Fig. 39), this again signifies the durability of our pillared membrane. Taken together, the high water permeability and selectivity outperform the widely reported 2D membranes (Supplementary Fig. 43, Supplementary Table 3), and the tunable water permeation behavior in aggressive environments promises a robust ACN membrane for real-world desalination.

## Discussion

We showed that the chemically inert CN, as a recognized challenge for membrane assembly, can be engineered to afford a real 2D morphology with an activated subnanochannel. The conformally packed membrane fostered by polycation clusters is then presented with a noncovalent but well-interlocked sandwiched nanostructure to withstand swelling in various aqueous environments. Unlike the disordered packing mode of conventional CN with ubiquitous structure defects, the regular transport passage of our pillared membrane allows both high water flux and salt rejection, which also survives in aggressive conditions and therefore expands its applicability. We envision that such a stable lamellar structure with a nanoconfinement effect may qualify its further uses in membrane reactors which integrate separation and catalytic ability if the semiconductor properties of photoactive CN are fully exploited to regulate mass and electron transfer pathways.

## Methods
### Chemicals and reagents
All chemicals were used as received without further purification. Melamine (C$_3$H$_6$N$_6$, 99%) was purchased from Alfa Aesar. Sucrose (C$_{12}$H$_{22}$O$_{11}$, BioXtra, ≥99.5%), aluminum chloride hexahydrate (AlCl$_3$·6H$_2$O, 99%), Ferron (8-hydroxy-7-iodo-5-quinolinesulfonic acid, ≥98.5%), 3-hydroxytyramine hydrochloride (99%), tris-base (C$_4$H$_{11}$NO$_3$, ≥ 99%), and concentrated hydrochloride (HCl, 37 wt%) were purchased from Sigma–Aldrich. Sodium hydroxide (NaOH, 98.6%) were purchased from VWR Chemicals. Silver nitrate (AgNO$_3$, ≥98.5%), sodium chloride (NaCl, ≥99.5%), lithium chloride anhydrous (LiCl, ≥99.5%), potassium chloride (KCl, ≥99%), calcium chloride dihydrate (CaCl$_2$·2H$_2$O, ≥99%), magnesium chloride (MgCl$_2$·6H$_2$O, 99%), acetone and methanol were purchased from Fischer Chemical. Sodium acetate (CH$_3$COONa), 1,10-phenanthroline (C$_{12}$H$_8$N$_2$, 99 + %), and hydroxylamine hydrochloride (NH$_2$OHHCl) were purchased from Fluka Analytical. Milli-Q water (conductivity = 0.055 μS/cm, Q3 level) with pH of 5.7 was used throughout the experiments. Characterizations are listed in Supplementary Methods.

### Preparation of polymeric carbon nitride (CN) and CN nanosheet
Melamine (5 g) was loaded into a porcelain crucible covered with a lid, which was then transferred into a muffle furnace and heated to 550 °C for 4 h with a ramping rate of 2.3 °C min$^{-1}$ in air atmosphere, followed by cooling down to room temperature. The yellow product was ground into a fine powder and denoted as bulk CN. Then, 200 mg CN was dispersed in 200 mL H$_2$O, followed by continuous sonication for 12 h in an ultrasonic bath (Fisherbrand, 90 W) with full amplitude. The resulting suspension was centrifuged at 2375 × g for 10 min to remove large unexfoliated particles, upon which the supernatant was collected and further subjected to 3-day free standing for purification. Finally, the supernatant was extracted from the precipitate using a pipette and denoted as CN nanosheet.

### Preparation and analyses of Keggin Al$_{13}$ and Al$_{30}$
Keggin Al$_{13}$ and Al$_{30}$ were synthesized according to established procedures[38]. Specifically, 0.6 M NaOH solution (80 mL) was dropwise (~1 mL min$^{-1}$) added into 1.0 M AlCl$_3$·6H$_2$O solution (20 mL) under vigorous stirring at 60 °C (for Al$_{13}$) or 95 °C (for Al$_{30}$) in oil bath. The

resulting solution was continuously stirred for 12 h and then aged for 24 h at 60 °C (for $Al_{13}$) or 95 °C (for $Al_{30}$), upon which colorless solutions were obtained without precipitation. The final [OH]/[Al] molar ratio in $Al_{13}$ and $Al_{30}$ solutions was 2.4, and the concentration of Al was 0.2 M. Note that long-term aging may lead to the variation of Al species[61]; all the solutions were stored at room temperature for 5 days prior to further test and use.

To quantitatively determine the relative amounts of $Al_{13}$ and $Al_{30}$ solutions, Al-Ferron kinetics based on previous studies were evaluated in this work with slight modifications[62,63]. The Ferron reagent for colorimetric analysis consisted of three mixed solutions: (1) 500 mL aqueous solution containing $2.85 \times 10^{-3}$ M Ferron and $2.52 \times 10^{-4}$ M 1,10-phenanthroline, (2) 200 mL 4.3 M $CH_3COONa$ aqueous solution, (3) 200 mL acidified $NH_2OH \cdot HCl$ aqueous solution (containing 100 g $NH_2OH \cdot HCl$ and 40 $mL \cdot L^{-1}$ concentrated HCl. The three stocks were individually vacuum filtrated using a previously washed Nylon membrane filter (pore size: 0.45 μm) before taking them together. Solutions 2 and 3 were mixed prior to the addition of solution 1. The mixed solution was stored in the refrigerator (0–4 °C) in the dark and allowed for 1-week aging prior to further use.

Standard aluminum solution (0.01 M) was prepared by dissolving the salt $AlCl_3 \cdot 6H_2O$ in a known volume of 5 mM HCl, the molar ratio of HCl/Al was fixed at 0.5 to finally obtain a low pH value of ~2.3, which efficiently minimized Al hydrolysis. The standard $AlCl_3$, $Al_{13}$, and $Al_{30}$ solutions were diluted to $1 \times 10^{-4}$ M prior to mixing with Ferron reagent. Specifically, Ferron reagent and aluminum solution with a fixed molar ratio of 50 was transferred into 10 mm path-length quartz cuvette and placed into the chamber of a Shimadzu UV 2600 spectrometer. The kinetics scan was then immediately initiated (within 30 s), and absorbance changes were recorded at 363 nm with a reading frequency of 6 $min^{-1}$.

## Preparation of $Al_{30}$-CN and $Al_{13}$-CN powder

CN nanosheet supernatant was dropwise (1 mL $min^{-1}$) added into $Al_{13}$ or $Al_{30}$ solution at different [$Al_{13}$]/[CN] or [$Al_{30}$]/[CN] mass ratios (0.25:1, 1:1, 2:1, and 4:1, unless otherwise specified) under vigorous stirring at 60 °C (for $Al_{13}$) or 95 °C (for $Al_{30}$) in oil bath. The resulting mixture was continuously stirred for 12 h and then aged for 24 h at 60 °C (for $Al_{13}$-CN) or 95 °C (for $Al_{30}$-CN). The as-obtained suspension was centrifuged at $7700 \times g$ for 15 min to collect the precipitate, which was then dispersed in water and washed by vacuum filtration, 0.5 M $AgNO_3$ solution was used to detect any residual chloride species in the filtrate. The sample was finally dried under vacuum at different temperatures (20, 60, and 100 °C) for >15 h. For comparison purposes, the CN nanosheet was also aged under the same condition (95 °C), which was then collected by centrifugation and dried for further analysis.

## Preparation of CN and $Al_{30}$-CN membranes

(1) Polydopamine (PDA)-coated polyether sulfone (PES) membrane filter: Tris-base (0.1 M) was dropwise added with 0.1 M HCl to afford basic tris-HCl solution (10 mM, pH = 8.5). Dopamine chloride (2 mg $mL^{-1}$) was dissolved in tris-HCl via gentle stirring at room temperature. PES (Sterlitech, USA, diameter: 25 mm, pore size: 0.2 μm, thickness: 125 μm) membrane filter was immediately immersed in the solution and kept for 24 h, upon which PES surface was coated with PDA. Subsequently, the gray PDA-PES filters were thoroughly rinsed with water and baked overnight at 60 °C under a vacuum prior to further use. Note that the bare PDA-PES filter shows no water flux but high salt permeation rate (3.1 mol $m^{-2}$ $h^{-1}$) when applied in forward osmosis, which was then considered to exert no impact on the evaluation of membrane performances.

(2) CN membrane: The CN nanosheet suspension was further centrifuged at $7700 \times g$ for 15 min to collect the supernatant, which

was then diluted and vacuum filtrated onto PDA-PES filter. Given that thinner membrane (thickness < 300 nm) potentially leads to the formation of pin holes or incompletely covered regions on PDA-PES filter, CN membranes were prepared with thicknesses of 350, 500, 740, and 1500 nm in this work (Supplementary Fig. 16a). The membranes were dried at 60 °C under vacuum at room temperature for > 15 h, unless otherwise mentioned.

(3) $Al_{30}$-CN (abbreviated as ACN) membrane: The $Al_{30}$-CN suspension (0.25:1, 1:1, 2:1, and 4:1, hereafter referred to ACN-1, ACN-2, ACN-3, and ACN-4, respectively) after aging was centrifuged at $7700 \times g$ for 15 min to collect the supernatant, which was then diluted and vacuum filtrated onto PDA-PES filter with the desired thickness. Subsequently, the membrane clamped on the vacuum suction device was in situ washed with water to remove residual chloride (examined by 0.5 M $AgNO_3$ aqueous solution). The washed ACN membranes were finally dried under vacuum at varied temperatures (20, 60, and 100 °C) for >15 h (thickness: 320, 500, 720, and 1500 nm, Supplementary Fig. 16b).

## Membrane stability test

The ACN membrane dried at 60 °C under a vacuum was immersed in water for 24 h at room temperature, which was then immediately subjected to XRD test after quickly wiping off the liquid drops on the membrane surface. Subsequently, the membrane was dried again at 60 °C under vacuum for 24 h. The drying-wetting process was conducted for three cycles to evaluate the stability of ACN membranes in water. Note that the membranes dried at 20 °C under vacuum were soon detached from PDA-PES substrate and disintegrated upon contact with water, but the membranes dried at 60 °C and 100 °C under vacuum were rather stable in water to resist disintegration, even after soaking for one month. The ACN membrane dried at 60 °C under vacuum was also immersed in various environments for 24 h, including aqueous salt solutions (0.2 M, KCl, NaCl, KCl, $CaCl_2$, and $MgCl_2$), acidic (pH = 2) and alkaline (pH = 12) solutions, to evaluate its adaptive stability.

## Ion permeation tests of membranes

A customized H-shaped cell with two compartments was applied for ion permeation tests, as shown in Supplementary Fig. 26a. The membrane facing the feed side was sealed in between two hollow silicon pads with an opening of 1 cm in diameter, which was then clamped between two compartments to afford a leak-free environment for reliable permeation tests. The effective membrane area was 0.785 $cm^2$ in this work. The feed and permeate reservoirs were filled with 50 mL aqueous salt solutions (0.2 M KCl, NaCl, KCl, $CaCl_2$, or $MgCl_2$, unless otherwise mentioned) and equivalent water, respectively. During the permeation test, the two sides were magnetically stirred to minimize concentration polarization. The concentration variation at permeate side was monitored by a conductivity meter (MultiLab 540) to calculate the ion permeation rate ($P$) using the following equations:

$$\lambda = \Delta C \times \Lambda_m \quad (1)$$

$$P = \Delta C \times V / (A \times \Delta t) \quad (2)$$

where $\lambda$ is the measured ion conductivity at permeate side, $\Lambda_m$ is the molar conductivity, $\Delta C$ is the liquid concentration at permeate side, $V$ is the volume of solution at permeate side, $A$ is the effective membrane area, and $\Delta t$ is the permeation time (24 h in this work).

## Water flux and salt rejection tests of membranes

Water flux and salt rejection tests were carried out in forward osmosis mode with a similar H-shaped cell configuration (Supplementary Fig. 26b). The feed and draw compartments were filled with 0.1 M NaCl solution and 2 M sucrose solution, respectively. The osmotic pressure

generated by 2 M sucrose was estimated to be 49 bar (at room temperature) according to van't Hoff equation:

$$\pi = c \times R \times T \qquad (3)$$

where $c$ is the molarity of the solution, $R$ is the gas constant (0.08206 L atm mol$^{-1}$ K$^{-1}$), $T$ is the Kelvin temperature. The liquids in both compartments were also magnetically stirred during the test. The water flux ($J$) can be calculated according to the following equation:

$$J = \Delta V / (A \times \Delta t) \qquad (4)$$

where $\Delta V$ is indicated by the liquid height change over time ($\Delta t$) at the draw side, $A$ is the effective membrane area.

Salt rejection ($R$) can be calculated based on the following equation:

$$R = 1 - C_d / C_f \qquad (5)$$

where $C_d$ and $C_f$ denote the salt concentration at the draw and feed sides, respectively.

The initial pH value of Milli-Q water was measured to be 5.7 in this work, which was then defined as "neutral" water. For the water flux and salt rejection tests of ACN membrane under acidic and alkaline conditions, the pH values of aqueous NaCl solution at the feed side were adjusted to 2 and 12 by 1 M HCl or NaOH solutions, respectively. In this case, the concentration of Na$^+$ at the draw side was quantitatively determined by inductively coupled plasma-optical emission spectrometry (ICP-OES).

### Computational details

Density functional theory (DFT) calculations were carried out using CP2K quantum chemistry software package[64]; the nonlocal exchange and correlation function in Hamiltonian were described by Perdew-Burke-Ernzerhof (PBE) parametrization based on the generalized gradient approximation (GGA). The van der Waals (vdW) interactions were taken into account using the DFT-D3 method proposed by Grimme[65]. The GTH potential and Molopt basis set (DZVP-MOLOPT-SR-GTH) were adopted with an energy cutoff of 400 Ry[66]. In terms of the geometric configuration optimization, both atomic position and cell parameters were relaxed, with the maximum force being lower than $4.5 \times 10^{-4}$ Ha/bohr. A spin-polarized scheme was employed for all the calculations. The adsorption energy ($E_a$) was calculated using the following equation:

$$E_a = E_{(CN+Keggin)} - E_{CN} - E_{Keggin} \qquad (6)$$

where $E_{(CN+Keggin)}$ and $E_{CN}$ are the total energies of the CN nanosheet (model in Supplementary Fig. 5) with and without adsorbed Keggin ions, respectively; $E_{Keggin}$ is the energy of adsorbed Keggin ions.

The calculation of Fukui index was performed by the following equations:

$$f_+(r) = \rho_{N+1}(r) - \rho_N(r) \qquad (7)$$

$$f_-(r) = \rho_N(r) - \rho_{N-1}(r) \qquad (8)$$

Note that some of the studies adopted a perfectly stacked CN model for DFT calculation, in which the graphitic carbon nitride comprises an ideal tri-s-triazine-based 2D covalent bond network, without structural defects. Such a hypothesized model, however, fails to represent the real structure of CN. Practically, the imperfect 2D CN framework (termed polymeric carbon nitride) composed of zig-zag

tri-s-triazine-based chains connected via intramolecular hydrogen bonds can better reveal the in-plane staking mode.

## Data availability

All data generated or analyzed during this study are included in this published article and its supplementary information files.

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

## Acknowledgements

Y.W. thanks the Alexander von Humboldt Foundation for a postdoctoral fellowship. This work was financially supported by the Max Planck Society.

## Author contributions

Y.W. and M.A. conceived the project. Y.W. synthesized the precursors, fabricated the membranes, and carried out the permeation tests. T.L. helped in the ion permeation measurement. N.V.T. contributed to electron microscopy experiments and data processing. J.Y. analyzed membrane structure and permeation results. Y.W. and M.A. analyzed the data and wrote the manuscript with input from all authors. All authors discussed the results and commented on the manuscript.

## Funding

## Competing interests

The authors declare no competing interests.
