## [Peer Review File · Nature Communications]

Lamellar carbon nitride membrane for enhanced ion sieving and water desalinationREVIEWER COMMENTS

Reviewer #1 (Remarks to the Author):

The transport mechanism lacks more detailed characterization, for this reason, I invite the authors to carry out an ion permeation as a function of temperature to find out the energy barriers needed to cross the membrane barrier. Is the transport of water molecules happening only in a zig-zag configuration or goes also through the pores on the ACN? Are these pores free or occluded? More characterization of the nanosheets should be highlighted

One of the key points of the paper is that, contrary to other 2D-based membranes, the membrane does not disintegrate when heated at 60 or 100-degree celsius. However, the long-term stability test carried out only for 6h poses a serious question about the feasibility of these potential FO membranes since they should last an order of magnitude longer in time. I would suggest the authors to report a stability test even only for one single membrane (ACN3) for a longer time.

The authors claim that the membrane expands and contracts based on the PXRD experiment carried out in Figure 3e. I believe that such a statement is too strong based on the reported data: the d_{spacing} ($\Delta 2\theta < 0.8^\circ$) variation is very small (as expected) and relies on ex-situ wetting/drying experiments to hydrate/dehydrate the membrane and re-run the XRD experiment could yield shifts in 2θ due to z-alignment of the sample. Moreover, the top blue curve belonging to the "dry" state is shifted in comparison to the bottom "dried-sample" curves. Unless an in-situ measurement is performed (or a higher intensity beam is used) I would not accept this data for their claim.

Please specify on the panel figure that the 5.6 and 5.9 angstroms refer to the gallery spacing (free spacing). It might get confusing for the reader since it seems that the peaks centered at $15.0^\circ < 2\theta < 15.8^\circ$ have d_{spacing} of 5.6 and 5.9 angstrom, respectively.

The authors provide an HR-TEM micrograph for the intercalated polycation and show the d-spacing that seems to be in good agreement with XRD. They further show the same micrograph for pure ACN and no fringes are noted here. Can the authors provide electron diffraction for both samples to measure the d-spacing?

Can the author explain why when they use Al13 instead of Al30 there is no new peak showing a new ordered configuration of the nanosheets? Do they believe that it has to do only with the different adsorption energies? The reader would appreciate it if the nominal geometric dimensions of Al13 were reported too.

Can the authors show a cross-section of the membranes that were intercalated with Al13? Ideally, they should try to measure its performance as well: if the XRD (Fig. 6 SI) did not show a peak of periodically arranged nanosheets along the z-axis it could always be that there is a short-order range and that the XRD cannot pick it up.

The authors report that their membranes could be used in FO desalination. However, can they explain how their fabrication method (vacuum filtration) can be scaled to a large, industrial scale and how to ensure uniform thickness (700 nm) on this scale avoiding thick-coated regions as well as uncoated ones? I believe that the coexistence of thick and thin coated regions generates cracks in the film. Can the authors mention what happens to the membrane film when the thickness goes beyond 1500 nm, especially for the ACNs membranes?

"Given the thickness of ideal single-layered CN (3.2 Å), the free spacing of ACN for molecules and ions to access is calculated as 6.6 Å, close to the size of hydrated monovalent salt ions. Al30 with nominal geometric dimensions of 10 Å × 10 Å × 20 Å³⁸, even lying flat in-between CN layers with optimal configurations (Supplementary Fig. 5), is assumed to afford larger free spacing of ~10 Å, but the loss of η-H₂O molecules during post-drying process can explain the smaller-than-expected value" Can the authors back this statement with data? Why the wetting/drying experiment does not show the full d-spacing extent (3.2+10 angstrom)?

The average size of the nanosheets should be reported with statistical analysis.

The authors should specify the filtration condition used to obtain their membranes and describe a usual filtration experiment where the concentration of the solution vs the obtained layer thickness should be also discussed.

I could not find the thickness of the PES support. The performance of your membranes could be even higher: In FO two types of concentration polarization are present. The external one goes from the bulk feed solution to the membrane or support (depending on which phase faces the feed solution), and an internal one, takes place within the porous support. While the former can be minimized to a certain extent with stirring, the latter becomes hard to overcome and it is a function of the support transport resistance. The osmotic pressure driving force is calculated based on the bulk concentration, while, due to polarization concentration, the real driving force is lower. It is worth investigating (for future studies) the same system with different supports and thicknesses.

A Robeson-type plot of Table S3 would become a very useful impact in highlighting the performance of the membranes.

Line 177. The potassium cation needs the superscript to be so.

Figure 4a. The minor ticks in the y-axis are absent between 10⁻³ and 10⁻².

Reviewer #2 (Remarks to the Author):

This manuscript introduced a CNT/PVA film as a substrate with the aim to increase mechanical and chemical cleaning properties of the prepared TFC NF membrane. The paper is well-organized. However, some revisions are required.

1. CN membrane and ACN membrane stability: Authors have stated that the ACN membrane is quite stable when immersed in water for 24 h. How is the stability of the fabricated ACN membranes at a longer operation time? Authors have tested their fabricated ACN / CN membranes for just only 6 hours of operating times.
2. CN membrane and ACN membrane preparation: What is the purpose of coating PES-based membrane filter with the polydopamine (PDA)? Could the PDA form a non-covalent interaction with the CN or ACN?

Reviewer #3 (Remarks to the Author):

The manuscript reported the preparation of carbon nitride membrane, which was used as an ion sieving and water desalination. However, I have some questions related to the content and it has some typos therefore I suggest the manuscript required major revision to get published.

1. Analyzing the Membrane-based water treatment, the quantity of g-CN determine the activity of the synthesized Membrane. So the authors should investigate the influence of immersion g-CN on the Membrane activity.
2. To demonstrate the critical role of the carbon nitride, the achieving salt rejection and water flux performance of bare Membrane should be studied.
3. In explaining the successful functionalization of Membrane, it is required to draw graphs and compare them with the same carbon nitride based membrane in the literature (for example reported papers:
J. Mater. Chem. A, 2020,8, 19133-19155. Sep. Purif. Technol., 2019, 215, 430-440. Science of The Total Environment Volume 792, 20 October 2021, 148462 Purif. Technol., 2020, 235, 116134. Ind. Eng. Chem. Res. 2021, 60, 25, 9189–9195. Water Res 2021 Jul 15;200:117207.

10.1016/j.cjche.2021.01.011

4. Another critical issue is that it is unknown how much g-CN and Al were grafted on the membrane?
5. the N₂ adsorption, the surface area and pore volume missed totally in this manuscript.

Response to Reviewers' Comments

Reviewer #1 (Remarks to the Author):

Response: The authors appreciate the valuable comments from Reviewer #1, the detailed responses are listed as follows.

1. The transport mechanism lacks more detailed characterization, for this reason, I invite the authors to carry out an ion permeation as a function of temperature to find out the energy barriers needed to cross the membrane barrier. Is the transport of water molecules happening only in a zig-zag configuration or goes also through the pores on the ACN? Are these pores free or occluded? More characterization of the nanosheets should be highlighted.

Response: Reviewer #1 raised a good point. Accordingly, we carried out the permeation experiments at different temperatures to explore the mechanism of ion permeation through the membranes. As shown in Fig. R1, the permeation rates of these ions follow the Arrhenius equation, $\exp(-E/k_B T)$, where E is the energy barrier and k_B is the Boltzmann constant. One can see that the energy barrier for monovalent and divalent ions are 11.5-13.7 kJ mol⁻¹ and 41.8-44.8 kJ mol⁻¹, respectively (inset of Fig. R1). Compared with monovalent ions, it turns to be more difficult for divalent ions to permeate through our membrane, this keeps good consistency with the permeation rates shown in Fig. 4b in main text.

Fig. R1 Temperature dependent ion permeation of ACN-3 membranes. Dotted lines are best fits to the Arrhenius behavior. Inset: energy barriers for various ions.

We can explain this phenomenon in more details: One can gain the common knowledge that the higher ion charge, the stronger it attracts water molecules. As such, the ions with higher hydration free energies (as listed in Supplementary Table 2) are more prone to experience larger barriers for entering into the subnanometer channels and exponentially lower permeation rates. Our supplemented data supports that the energy barrier associated with dehydration is the dominant effect in our case of subnanometer channels.

We have added the experimental results into the revised manuscript and updated as new Supplementary Fig. 29. Accordingly, we explained the ion permeation behavior in the main text.

Page 5, Lines 194-201:

“In comparison, ACN-3 membrane quantifies substantially decreased permeation rates at levels of $10^{-3} \text{ mol m}^{-2} \text{ h}^{-1}$ for K^+ , Li^+ and of $10^{-4} \text{ mol m}^{-2} \text{ h}^{-1}$ for Ca^{2+} and Mg^{2+} (Fig. 4b). That is, the ion permeation behavior is governed by size-exclusion effect when the activated subnanochannel shows narrower free spacing than the hydrated diameters of these cations (Supplementary Table 2). The lower permeation rate of divalent species relative to the monovalent ones is ascribed to the higher dehydration energy barrier^{43,44}. This is further supported by the temperature dependent ion permeation behavior (Supplementary Fig. 29), which follows the Arrhenius equation, $\exp(-E/k_B T)$, where E is the energy barrier and k_B is the Boltzmann constant⁴⁴. The E values of divalent ions (41.8 kJ mol^{-1} for Ca^{2+} and 44.8 kJ mol^{-1} for Mg^{2+}) are much higher than that of monovalent ones ($11.5, 13.7$ and 13.1 kJ mol^{-1} for K^+ , Na^+ and Li^+ , respectively). As such, the divalent species are less likely to enter into the subnanometer-scale nanochannels and therefore coming with exponentially lower permeation rates.”

We interpret the “zig-zag configuration” described by Reviewer #1 as the subnanochannels indicated by the arrows in Fig. 1b in main text, which constitutes the primary transport passage in this work. When it comes to the possibility of water molecules permeating through the intraplanar pores, we would like to highlight some structural characteristics of polymeric CN (also widely termed as g- C_3N_4 in literature) which also hold true for ACN:

The intralayer pores, as schemed in various publications, can only account for a larger proportion in an ideally stacked crystalline structure, such as crystalline CN comprising imide-linked triazines (poly(triazine) imide), abbreviated as PTI).^[R1-R3] Only in this phase, the triangular pores can be visualized by TEM. In contrast, the extensively studied polymeric CN is predominantly characterized by the intraplanar hydrogen-bonded tri-s-triazine frameworks (structural model in Supplementary Fig. 5), instead of the triangular pores that formed via rigid covalent C-N bonding^[R4]. As such, the water molecules are less likely to transport through the “pores” in both CN and ACN. Assuming that there exist some “pores”, they can be largely occluded, because the nanosheets are not single layers (3-4 layers) and the stacking fault can make them not accessible for foreign molecules. In this work, pillared by the polycations, the activated subnanochannel in

ACN shows evidently higher water permeation rates than that of CN, suggesting the dominant pathway of “zig-zag configuration” described by Reviewer #1.

We understand the concern that it would be desirable to characterize these nanosheets to probe whether they contain intraplanar pores. It is however not possible for the whole scientific community working on polymeric CN, because it does not follow a rigid covalent bonding and is presented with semi (or weak)-crystalline structure, as discussed above. The above-mentioned PTI may serve as an alternative, but it is only composed of small crystallites instead of nanosheets, making it difficult in terms of membrane-formation ability. It also takes complicated and unsafe procedures to synthesize PTI, while polymeric CN combines the merits of easy preparation and processing.

As suggested by Reviewer #1, we should convey the information to our readers. In this regard, we supplemented the discussion with some key points to clarify the observed phenomenon in terms of water permeation pathway.

Page 6, Lines 222-232:

“Comparatively, our ACN membranes are capable of facilitating water transport, as evidenced from the steady water flux with prolonging permeation time (inset of Fig. 4c, Supplementary Figs. 35-36). Note that both CN and ACN are structurally connected by intraplanar hydrogen bonds to bridge the tri-s-triazine motifs (Supplementary Fig. 5), they are less likely to form intralayer pores that can allow water molecules to fit in properly. These “pores” can only be formed by rigid covalent C-N bonding of ideally stacked graphitic CN (also widely termed as g-C₃N₄), which have not been experimentally available to date³³. Although it is seemingly possible to apply poly(triazine) imide (PTI) with regular triangular intralayer pores, the synthesized PTI is only presented with small crystallites rather than nanosheets, largely restricting their membrane-formation ability for separation purposes⁴⁶⁻⁴⁸. In this regard, the evidently enhanced water permeation of ACN membrane can be primarily ascribed to the regular subnanochannel activated by Al₃₀”.

[R1] *Angew. Chem. Int. Ed.* **61**, e202207457 (2022).

[R2] *Sci. Adv.* **6**, eabb6011 (2020).

[R3] *Adv. Mater.* **33**, 2106359 (2021).

[R4] *Nat. Rev. Mater.* **2**, 17030 (2017).

2. One of the key points of the paper is that, contrary to other 2D-based membranes, the membrane does not disintegrate when heated at 60 or 100-degree celsius. However, the long-term stability test carried out only for 6h poses a serious question about the feasibility of these potential FO membranes since they should last an order of magnitude longer in time. I would suggest the authors to report a stability test even only for one single membrane (ACN3) for a longer time.

Response: In previous manuscript, we fixed the permeation time with a duration of 6 hours to simply compare the performance with some of the recently reported cases in FO mode (3-8 hours)^[R5-R8]. Following the suggestion from Reviewer #1, we investigated the feasibility of ACN-3 membranes by prolonging the permeation time to 72 h (3 days), which were also studied in acidic and alkaline solutions. As shown in Fig. R2, our membrane only experienced slight decrease of water flux (within 10%) and salt rejection (within 2.8%) in these environments, suggesting the favorable long-term feasibility of our membrane for potential FO applications.

Accordingly, the supplemented data has been added and plotted as updated Supplementary Fig. 39. The experimental results are also described in main text as follows.

Page 6, Lines 239-242:

“Upon prolonging the permeation time up to 72 hours, we found that the ACN-3 membrane only experiences slight drop of water flux and salt rejection, suggesting the favorable long-term feasibility that can potentially allow its use in forward osmosis (Supplementary Fig. 39).”

Page 7, Lines 287-289:

“Long-term operation of the ACN-3 membrane in acidic and alkaline environments for 72 hours only results in a decrease of water flux within 10% and salt rejection within 2.8% (Supplementary Fig. 39), this again signifies the durability of our membrane”.

Fig. R2 Long-term stability test of ACN-3 membrane operated in acidic, neutral and alkaline solutions. Water flux and salt rejection are plotted as a function of permeation time up to 72 hours.

[R5] *Nature* **550**, 380-383 (2017).

[R6] *Nat. Sustain.* **3**, 296-302 (2020).

[R7] *ACS Nano* **11**, 11082-11090 (2017).

[R8] *J. Am. Chem. Soc.* **143**, 5080-5090 (2021).

3. The authors claim that the membrane expands and contracts based on the PXRD experiment carried out in Figure 3e. I believe that such a statement is too strong based on the reported data: the d-spacing ($\Delta 2\theta < 0.8^\circ$) variation is very small (as expected) and relies on ex-situ wetting/drying experiments to hydrate/dehydrate the membrane and re-run the XRD experiment could yield shifts in 2θ due to z-alignment of the sample. Moreover, the top blue curve belonging to the “dry” state is shifted in comparison to the bottom “dried-sample” curves. Unless an in-situ measurement is performed (or a higher intensity beam is used) I would not accept this data for their claim.

Response: We agree with the raised point in terms of the d-spacing change, which is only characterized by subtle fluctuations. We tried to carry out the in-situ measurement or using a higher intensity beam, but the thin film with semi- or weak-crystalline nature prevents us from getting better result. As suggested by Reviewer #1, the statement needs to be toned down and should point to the “fluctuation” instead of precise expansion or contraction. In this regard, we deleted the arrows in Fig. 3e to avoid misunderstanding, and interpreted the results as follows in main text.

Page 4, lines 156-160:

“The conformally packed membrane baked at 60 °C shows excellent swelling resistance when immersed in water and aqueous salt solutions, as evidenced by the minimal change of free spacing (Fig. 3e, Supplementary Figs. 19-25, Supplementary Note 3) in wet or dry state, which fluctuates within a narrow range of 5.6-5.9 Å”.

4. Please specify on the panel figure that the 5.6 and 5.9 angstroms refer to the gallery spacing (free spacing). It might get confusing for the reader since it seems that the peaks centered at $15.0^\circ < 2\theta < 15.8^\circ$ have d_spacing of 5.6 and 5.9 angstrom, respectively.

Response: The authors appreciate Reviewer #1 for careful correction. We revised this panel figure and its accompanying legend in main text, which is shown as follows in more details:

“e, XRD patterns of ACN-3 membranes subjected to drying-wetting cycles, each drying or wetting event lasts for 24 hours. The grey region marked in e represents the free spacing fluctuating within 5.6-5.9 Å in dry or wet state, the broad diffraction peak centered within $2\theta = 16-20^\circ$ is assigned to the polydopamine-coated polyether sulfone filter.”

5. The authors provide an HR-TEM micrograph for the intercalated polycation and show the d-spacing that seems to be in good agreement with XRD. They further show the same micrograph for pure ACN and no fringes are noted here. Can the authors provide electron diffraction for both samples to measure the d-spacing?

Response: The TEM image and d-spacing shown in Fig. 2b belongs to ACN, not the intercalated polycation itself. As mentioned in the “Methods” section, the obtained polycations are colorless solutions, which have been characterized by NMR and kinetics studies (Supplementary Figs. 2 and 3). When they are used to intercalate into CN layers, the final ACN is characterized by identifiable fringes (Fig. 2b) in HR-TEM images, which is not observed in pristine CN without intercalation (Supplementary Fig. S12). Following the suggestion from Reviewer #1, we recorded the electron diffraction images for both ACN and CN (Fig. R3). One can see that the diffraction spots of ACN is identifiable (although presented with diffuse scattering), upon which the d-spacing is calculated to be around 9.9 Å, keeping in line with the XRD and TEM results shown in

main text. In contrast, the nearly amorphous structure of CN points to a poor stacking behavior. Although the diffraction peak of pristine CN centered at 27.5° is shown in XRD result, the inherently less ordered structure makes it unstable when approached by electron beam. Such a comparison highlights the improved and conformal stacking of ACN, enabled by introducing polycations as pillaring agents.

We would like to appreciate Reviewer #1 for this suggestion to reveal our materials in more details, which have also been added as updated Supplementary Fig. 12.

Fig. R3 Selected area electron diffraction images. a, ACN-3, b, CN. The scale bars in **a** and **b** are 1 nm^{-1} . The arrow in **a** denotes an interlayer distance of around 9.9 \AA .

6. Can the author explain why when they use Al_{13} instead of Al_{30} there is no new peak showing a new ordered configuration of the nanosheets? Do they believe that it has to do only with the different adsorption energies? The reader would appreciate it if the nominal geometric dimensions of Al_{13} were reported too.

Response: The adsorption energy differences were revealed by DFT calculations. Theoretically, Al_{13} with smaller nominal geometric dimensions of $9 \text{ \AA} \times 9 \text{ \AA} \times 9 \text{ \AA}$, is more likely to intercalate into CN layers due to its lower steric hinderance. However, as discussed in Supplementary Note 1, the acidity and charge of Al_{30} are much higher than that of Al_{13} , which can interact intensively with the abundant basic sites of CN by acid-base interaction (more specifically, CN can be effectively protonated by Al_{30}), this is also reflected by the zeta potential changes (Supplementary Figs. 6-7). Taking together, these experimental differences are in accordance with the calculation results.

We have added the nominal geometric dimensions of Al_{13} into the main text.

Page 3, Lines 106-109:

“The high-charge Al_{30} , characterized by asymmetric distribution of acidic sites³⁸ and highest acidity centered at the equatorial region (Fig. 1, Supplementary Fig. 1, sites 3 and 4 of bound water, η - H_2O), is found to interact with CN at much higher adsorption energy than the lower acidic $[Al_{13}O_4(OH)_{24}(H_2O)_{12}]^{7+}$ cluster (abbreviated as Al_{13}) despite its smaller size ($9 \text{ \AA} \times 9 \text{ \AA} \times 9 \text{ \AA}$)³⁹ and lower steric hinderance (Supplementary Figs. 1-11, Supplementary Note 1).”

7. Can the authors show a cross-section of the membranes that were intercalated with Al_{13} ? Ideally, they should try to measure its performance as well: if the XRD (Fig. 6 SI) did not show a peak of periodically arranged nanosheets along the z-axis it could always be that there is a short-order range and that the XRD cannot pick it up.

Response: Below we provide the cross-sectional view of Al_{13} -CN membrane (Fig. R4a), which is also presented with particle-like staking behavior, highly resembling that of pristine CN membrane. Also, we investigated the ion permeation behavior using Al_{13} -CN membrane (Fig. R4b). By comparing these results with CN and Al_{30} -CN (which was abbreviated as ACN throughout the paper) membranes, it is evident that Al_{13} -CN membrane only gives comparable ion permeation performance with that of CN, suggesting the nonselective transport channel due to the absence of well-stacked layers when intercalated by Al_{13} .

We added these results as new Supplementary Fig. 28 and described in main text.

Page 5, Lines 187-189:

“Similarly, the Al_{13} -CN membranes (thickness: 750 nm) also show comparable permeation rates of these ions with that of CN ones (Supplementary Fig. 28), due to the absence of activated transport channels.”

Fig. R4 Microstructure and ion permeation behavior of Al_{13} -CN membrane (thickness: ~750 nm). **a**, Cross-sectional SEM image of Al_{13} -CN membrane (scale bar: 1 μ m), **b**, Comparison of the ion permeation performance of CN, Al_{13} -CN and ACN membranes.

Following the suggestion from Reviewer #1 that a short-order range shall exist but may not be revealed by XRD, we recorded high-resolution TEM and electron diffraction images of Al_{13} -CN. As

shown in Fig. R5, no diffraction pattern can be detected (resembles the case in pristine CN), again suggesting the non-successful intercalation using Al_{13} .

Collectively, we have added these images as new Supplementary Fig. 12 and mentioned them in main text.

Page 4, Lines 119-120:

“Consistently, high-resolution transmission electron microscope (HR-TEM) images (Fig. 2b) reveal the thin-layer morphology of ACN with resolvable fringe separation which corresponds to an interlayer spacing similar to that of X-ray diffraction (XRD) result, the ordered stacking mode is however not observed in conventional CN nanosheets and Al_{13} -CN ones (Supplementary Fig. 12).”

Fig. R5 Characterizations of Al_{13} -CN. **a**, TEM images, **b**, selected area electron diffraction image. The scale bars are 50 nm (left panel) and 10 nm (right panel) in **a**, and 1 nm^{-1} in **b**.

8. The authors report that their membranes could be used in FO desalination. However, can they explain how their fabrication method (vacuum filtration) can be scaled to a large, industrial scale and how to ensure uniform thickness (700 nm) on this scale avoiding thick-coated regions as well as uncoated ones? I believe that the coexistence of thick and thin coated regions generates cracks in the film. Can the authors mention what happens to the membrane film when the thickness goes beyond 1500 nm, especially for the ACNs membranes?

Response: Unlike the commercially available FO membranes (thin-film composite) fabricated by interfacial polymerization with scaling-up ability to afford square-meter-level products, vacuum filtration is the predominant fabrication method for the large 2D membrane (at cm^2 scale) family, which may produce defects due to the small area of nanosheet layers when they are subjected to stacking at m^2 scale, as summarized and outlooked in recent reviews^[R9-R11]. Indeed, this constitutes one of the main challenges facing the 2D membrane community. In this regard, full understanding and control of the origin of defects (*e.g.* cracks) during channel synthesis and membrane-fabrication processes is required to the realization of membranes at industrial scale, for instance, using mixed-matrix membranes (by dispersing nanomaterials in continuous polymer matrix).

That being said, our pillared membrane shows some promising aspects. As indicated by the AFM study in Fig. 3, the pristine CN membrane with particle-like morphology gives both thick-coated and thin-coated regions (Fig. 3c), this arises from the random stacking of nanosheets under pressure (caused by vacuum filtration). In contrast, when pillared by polycations to afford conformal packing, the ACN membrane is quite uniform and comes with substantially alleviated thickness fluctuation (Fig. 3d). Reviewer #1 queries about the 1500-nm thick membrane, accordingly, we investigated both CN and ACN membranes using SEM and AFM. Again, we see similar situations with that of thinner membranes displayed in main text, this points to the fact our pillaring strategy is also effective in terms of fabricating thicker ACN membrane. Although it is now only available at lab scale, we would make joint efforts to alleviate the scaling-up dilemma in 2D membranes (Graphene, MXene, etc.) in future work.

The microstructural images of thicker membranes have been added as new Supplementary Fig. 18. Also, we have added information in main text to make this point clearer.

Page 4, Lines 147-152:

“The heterogeneity reflected by thick- and thin-coated regions can potentially generate cracks when CN membranes are scaled up. Enabled by polycation pillaring to provide conformal packing, the lamellar ACN membrane comes with flat surface relative to that of CN, the latter fluctuating substantially with high roughness (Figs. 3c-3d). The heterogenous and homogenous coating behavior also holds true for thicker CN and ACN membranes (~ 1500 nm), respectively (Supplementary Fig. 18).”

Fig. R6 Microstructural characterization. Cross-sectional SEM images of (a) ACN-3 and (b) CN membrane with thickness of 1500 nm, scale bar: 2 μm . c, d, AFM images of (c) ACN-3 (scale bar: 10 μm) and (d) CN membranes (scale bar: 10 μm) with corresponding height profiles (inset).

[R9] *Nat. Rev. Mater.* **6**, 294-312 (2021).

[R10] *Science* **356**, eaab0530 (2017).

[R11] *Nat. Rev. Mater.* **1**, 16018 (2016).

9. “Given the thickness of ideal single-layered CN (3.2 Å), the free spacing of ACN for molecules and ions to access is calculated as 6.6 Å, close to the size of hydrated monovalent salt ions. Al₃₀ with nominal geometric dimensions of 10 Å × 10 Å × 20 Å³⁸, even lying flat in-between CN layers with optimal configurations (Supplementary Fig. 5), is assumed to afford larger free spacing of ~10 Å, but the loss of η-H₂O molecules during post-drying process can explain the smaller-than-expected value” Can the authors back this statement with data? Why the wetting/drying experiment does not show the full d-spacing extent (3.2+10 angstrom)?

Response: Previously, we reached the above statement by the collected information from Al₃₀ structure and experimental results. Firstly, the polycation Al₃₀ is characterized by “bound waters (termed as η-H₂O in main text)” residing on Al-O polyhedra (Supplementary Fig. 1). The FTIR and XRD results shown in Fig. 2e signify the loss of water molecules and gradually narrowed interlayer spacing. Upon immersing in water for a long period, the interlayer spacing is not expanded to 3.2+10 angstrom. This suggests that “lose of water molecules from ACN” is an irreversible process. Based on these considerations, we inferred that this is caused by the loss of bound water “η-H₂O”, because: if it is caused by the loss of free water (in aqueous solution), the interlayer spacing would be enlarged upon immersion, which is the cases for the “swelling” 2D membranes (graphene oxide, MXene, etc.) but not the case for our membrane. This arises from the essential difference between “bound water (in Al₃₀)” and “free water (in aqueous solution)”, which, unfortunately, cannot be experimentally distinguished, this is largely restricted by the fact that Al₃₀ is prepared and presented together with other less-polymerized oligomers or monomers in the colorless solution (Supplementary Fig. 2, in which Al₃₀ accounts for 70%).

Taking together, we have to admit “the loss of bound water” is not a rigorous statement. We have modified this statement in main text. Related studies which also identified the loss of water molecules and smaller interlayer spacing than expected, were also cited.

Page 4, Lines 122-126:

“Al₃₀ with nominal geometric dimensions of 10 Å × 10 Å × 20 Å³⁸, even lying flat in-between CN layers with optimal configurations (Supplementary Fig. 5), is assumed to afford larger free spacing of ~10 Å, but the loss of water molecules during post-drying process (Fig. 2e) can explain the smaller-than-expected value^{38-40”.}

10. The average size of the nanosheets should be reported with statistical analysis.

Response: Below (Fig. R7) we show the statistical analyses of the nanosheet size and height, which has also been added and updated as new Supplementary Fig. 8. Accordingly, the description in Supplementary Note 1 has been corrected as follows:

“statistical analyses of the nanosheet size and height suggest that the thickness of these CN nanosheets was around 1.1-1.3 nm, i.e., 3-4 layers (single-layer thickness: ~ 0.32 nm) with lateral size of ~ 300 nm (Supplementary Figs. 8c-8d).”

Fig. R7 Statistical analyses of nanosheets (n = 50). **a**, Lateral size, **b**, Height.

11. The authors should specify the filtration condition used to obtain their membranes and describe a usual filtration experiment where the concentration of the solution vs the obtained layer thickness should be also discussed.

Response: The membrane thickness can be controlled by changing the volume or concentration of the filtered suspension. In our case, we fixed the concentration and obtained the membranes by changing the volume of the nanosheet suspension. Fig. R8 shows the variation of membrane thickness as a function of the concentration of the filtered suspension, which has also been plotted as new Supplementary Fig. 16. In main text, we added the following statement.

Page 4, Lines 141-142:

“The membrane thickness can be controlled by changing the volume of the filtered suspension (Supplementary Fig. 16).”

Page 9, Lines 377-378 (“Methods” section):

“CN membranes were prepared with thickness of 350, 500, 740 and 1500 nm in this work (Supplementary Fig. 16a).”

Page 10, Lines 385-387 (“Methods” section):

“The washed ACN membranes were finally dried under vacuum at varied temperatures (20 °C, 60 °C, and 100 °C) for > 15 h (thickness: 320, 500, 720 and 1500 nm, Supplementary Fig. 16b).”

Fig. R8 Variation of membrane thickness as a function of the volume of filtrated suspension. a, CN membrane, b, ACN membrane.

12. I could not find the thickness of the PES support. The performance of your membranes could be even higher: In FO two types of concentration polarization are present. The external one goes from the bulk feed solution to the membrane or support (depending on which phase faces the feed solution), and an internal one, takes place within the porous support. While the former can be minimized to a certain extent with stirring, the latter becomes hard to overcome and it is a function of the support transport resistance. The osmotic pressure driving force is calculated based on the bulk concentration, while, due to polarization concentration, the real driving force is lower. It is worth investigating (for future studies) the same system with different supports and thicknesses.

Response: The authors appreciate the kind remind from Reviewer #1 on the missing information of PES support, which was 125 μm-thick (pore size: 0.2 μm) in this work. Taking the internal concentration polarization into consideration, we then used different supports with different thicknesses (Fig. R9). It was found that when the support thickness was decreased to 65 μm, the water flux of ACN-3 membrane (thickness: 500 nm) increased from 3.1 L m² h⁻¹ to 4.1 L m² h⁻¹ without sacrificing the salt rejection performance, such a tendency was also observed by using

mixed cellulose ester (MCE) support. To further explore whether the pore structure difference can affect the FO performance, we used the anodized alumina oxide (AAO) with cylindrical pores and found no obvious change. That is, the use of thinner support can alleviate the internal concentration polarization, and consequently, assure higher osmotic pressure to finally afford higher FO performance.

Taking together, the FO performance using thinner PES support was added and plotted as new Supplementary Fig. 38. Also, we added the relevant description in main text.

Page 6, Lines 237-239:

“At optimized conditions, the 500 nm-thick ACN-3 membrane holds both high water flux of $3.0 \text{ L m}^{-2} \text{ h}^{-1}$ and NaCl rejection rate of 99.4% (Fig. 4d). The water flux can be further improved using thinner PES filter to alleviate internal concentration gradient, without sacrificing the salt rejection performance (Supplementary Fig. 38).”

In “Methods” section, we added the thickness information of PES support.

Fig. R9 Cross-sectional SEM images for thickness calculation of different supports and FO performance of ACN-3 membrane. **a, b**, Polyethersulfone (PES), **c, d**, Mixed cellulose ester (MCE), **e, f**, anodized alumina oxide (AAO). **g**, FO performance of 500 nm-thick ACN-3 membrane on different supports. Scale bars in **a-d**, **e** and **f** are 100 μm, 20 μm and 2 μm, respectively.

13. A Robeson-type plot of Table S3 would become a very useful impact in highlighting the performance of the membranes.

Response: Following the suggestion from Reviewer #1, the performance comparison is shown in Fig. R10, which is also added and plotted as new Supplementary Fig. 43 to highlight the performance of our membrane. The Robeson-type plot (water permeability vs. water-salt permselectivity) requires the salt permeability data for graphing, which is however not available in most of the reported 2D membranes.

Fig. R10 Water permeability and NaCl rejection performance of 2D membranes in forward osmosis.

14. Line 177. The potassium cation needs the superscript to be so.

Response: We have corrected this typo in the revised manuscript.

15. Figure 4a. The minor ticks in the y-axis are absent between 10⁻³ and 10⁻².

Response: This may arise from the loss of resolution during image integration, we have updated Figure 4a with higher resolution.

Reviewer #2 (Remarks to the Author):

This manuscript introduced a CNT/PVA film as a substrate with the aim to increase mechanical

and chemical cleaning properties of the prepared TFC NF membrane. The paper is well-organized. However, some revisions are required.

Response: The authors appreciate the positive comments from Reviewer #2, below we provide point-by-point responses to address the raised concerns.

1. CN membrane and ACN membrane stability: Authors have stated that the ACN membrane is quite stable when immersed in water for 24 h. How is the stability of the fabricated ACN membranes at a longer operation time? Authors have tested their fabricated ACN / CN membranes for just only 6 hours of operating times.

Response: In previous manuscript, we fixed the permeation time with a duration of 6 hours to simply compare the performance with some of the recently reported cases in FO mode (3-8 hours)^[R5-R8]. Following the suggestion from Reviewer #2, we investigated the feasibility of ACN-3 membranes by prolonging the permeation time to 72 h (3 days), which were also studied in acidic and alkaline solutions. As shown in Fig. R1, our membrane only experienced slight decrease of water flux (within 10%) and salt rejection (within 2.8%) in these environments, suggesting the favorable long-term feasibility of our membrane for potential FO applications.

Accordingly, the supplemented data has been added and plotted as updated Supplementary Fig. 39. The experimental results are also described in the main text as follows:

Page 6, Lines 239-242:

“Upon prolonging the permeation time up to 72 hours, we found that the ACN-3 membrane only experiences slight drop of water flux and salt rejection, suggesting the favorable long-term feasibility that can potentially allow its use in forward osmosis (Supplementary Fig. 39).”

Page 7, Lines 287-289:

“Long-term operation of the ACN-3 membrane in acidic and alkaline environments for 72 hours only results in a decrease of both water flux within 10% and salt rejection within 2.8% (Supplementary Fig. 39), this again signifies the durability of our membrane”.

Fig. R1 Long-term stability test of ACN-3 membrane operated in acidic, neutral and alkaline solutions. Water flux and salt rejection are plotted as a function of permeation time up to 72 hours.

[R1] *Nature* **550**, 380-383 (2017).

[R2] *Nat. Sustain.* **3**, 296-302 (2020).

[R3] *ACS Nano* **11**, 11082-11090 (2017).

[R4] *J. Am. Chem. Soc.* **143**, 5080-5090 (2021).

2. CN membrane and ACN membrane preparation: What is the purpose of coating PES-based membrane filter with the polydopamine (PDA)? Could the PDA form a non-covalent interaction with the CN or ACN?

Response: Yes. Coating the filter with PDA has been a common strategy to enhance the adhesion with the membranes by virtue of non-covalent interaction^[R5-R8], this can be beneficial for long-term operation of these membranes under high pressure. One may query whether PDA coating will affect the performance of CN or ACN membrane. In this regard, we tested the water flux and salt permeation of both PES and PDA-PES in forward osmosis mode. However, the bare membrane shows no water flux but high salt permeation rate. This is reasonable because the PES filter we used in this work is characterized by a pore size of 200 nm, and the pores are not occluded after coating with PDA (Supplementary Fig. 15b). Such big pores are not expected to reject the salt species (with hydrated diameters lower than 1 nm), which permeate with much

higher rates (3.4 and 3.1 mol m⁻² h⁻¹ for PES and PDA-PES, respectively) than that of CN and ACN membranes (Fig. 4c in main text).

On the other hand, as we schemed in Supplementary Fig. 32, the osmotic pressure can only be created when the membrane is semipermeable, *i.e.*, allowing one species to transport while blocking the other one. When the bare membrane was used in forward osmosis, both salt solution (feed side) and sucrose solution (draw side) can freely diffuse through the membrane, and consequently, the transmembrane pressure is too low to drive the transport of water.

We added the above-described information in main text and “Methods” section to highlight the basic information of bare membrane.

Page 5, lines 171-172:

“Next, we evaluated the ion permeation behavior through the assembled membranes (supported by polydopamine-coated polyether sulfone filter to enhance the adhesion with membranes) in a customized H-shaped cell equipped with two reservoirs (Supplementary Fig. 26a).”

Page 9, Lines 370-373:

“Note that the bare PDA-PES filter shows no water flux but high salt permeation rate (3.1 mol m⁻² h⁻¹) when applied in forward osmosis, which does not interfere with the evaluation of membrane performances.”

[R5] *Nat. Nanotech.* **16**, 337-343 (2021).

[R6] *Nat. Commun.* **11**, 1097 (2020).

[R7] *Angew. Chem. Int. Ed.* **56**, 4662-4711 (2017).

[R8] *J. Am. Chem. Soc.* **135**, 17679-17682 (2013).

Reviewer #3 (Remarks to the Author):

The manuscript reported the preparation of carbon nitride membrane, which was used as an ion sieving and water desalination. However, I have some questions related to the content and it has some typos therefore I suggest the manuscript required major revision to get published.

Response: The authors appreciate the valuable comments from Reviewer #3, the detailed responses are listed as follows.

1. Analyzing the Membrane-based water treatment, the quantity of g-CN determine the activity

of the synthesized Membrane. So the authors should investigate the influence of immersion g-CN on the Membrane activity.

Response: Following the raised concern on the effect of immersed g-CN on membrane activity, we carried out the cycling test to evaluate the membrane activity upon immersion treatment. As shown in Fig. R1, after alternate immersion (for four cycles, each immersion treatment lasts for 12 h in water), the membrane activity can be well maintained.

We have added the results and plotted as new Supplementary Fig. 37, which has also been described in main text:

Page 6, Lines 234-237:

“At optimized conditions, the 500 nm-thick ACN-3 membrane holds both high water flux of $3.0 \text{ L m}^{-2} \text{ h}^{-1}$ and NaCl rejection rate of 99.4% (Fig. 4d), which can be well maintained after alternate immersion treatments (Supplementary Fig. 37).”

Fig. R1 Cycling test of 500 nm-thick ACN-3 membrane. Prior to next-cycle test, the membrane was immersed in H_2O for 12 h.

2. To demonstrate the critical role of the carbon nitride, the achieving salt rejection and water flux performance of bare Membrane should be studied.

Response: We tested the water flux and salt permeation of the bare PDA-PES filter, which however shows no water flux but high salt permeation rate when applied in forward osmosis. This is reasonable because the PES filter we used in this work is characterized by a pore size of 200 nm, and the pores are not occluded after coating with PDA (polydopamine), as shown in Supplementary Fig. 15b. Such big pores are not expected to reject the salt species (with hydrated diameters lower than 1 nm), which permeate with much higher rates ($3.1 \text{ mol m}^{-2} \text{ h}^{-1}$) than that of CN and ACN membranes (Fig. 4c in main text).

On the other hand, as we schemed in Supplementary Fig. 32, the osmotic pressure can only be created when the membrane is semipermeable, *i.e.*, allowing one to transport while blocking the other one. When the bare membrane was used in forward osmosis, both salt solution and sucrose solution can freely diffuse through the membrane, and consequently, the transmembrane pressure is too low to drive the transport of water.

We added the above-described information in “Methods” section to highlight the basic information of bare membrane.

Page 9, Lines 370-373:

“Note that the bare PDA-PES filter shows no water flux but high salt permeation rate ($3.1 \text{ mol m}^{-2} \text{ h}^{-1}$) when applied in forward osmosis, which does not interfere with the evaluation of membrane performances.”

3. In explaining the successful functionalization of Membrane, it is required to draw graphs and compare them with the same carbon nitride based membrane in the literature (for example reported papers: J. Mater. Chem. A, 2020,8, 19133-19155. Sep. Purif. Technol., 2019, 215, 430-440. Science of The Total Environment Volume 792, 20 October 2021, 148462 Purif. Technol., 2020, 235, 116134. Ind. Eng. Chem. Res. 2021, 60, 25, 9189–9195. Water Res 2021 Jul 15;200:117207. 10.1016/j.cjche.2021.01.011)

Response: We appreciate Reviewer #3 for listing the related references for comparison. However, the study on carbon nitride based membranes for forward osmosis remains quite limited. In very scarce cases^[R1,R2], they are only added as a filler or modifier to enhance the performance of commercial thin-film composite or thin-film nanocomposite, with missing data for a fair comparison of salt rejection performance. Taking these into consideration, we plotted the graph to rationally compare with other 2D membranes to highlight the improved performance in our case, as shown in Fig. R2.

This is also added and plotted as new Supplementary Fig. 43. Furthermore, recent reviews suggested by Reviewer #3 are cited for the benefit of readers (Highlighted in Reference list in main text).

Fig. R2 Water permeability and NaCl rejection performance of 2D membranes in forward osmosis.

4. Another critical issue is that it is unknown how much g-CN and Al were grated on the membrane?

Response: The CN and Al contents of these membranes were determined by Argon-ion sputtered XPS depth profile (Supplementary Fig. 24), their contents at each depth can be recorded to obtain average ratio values, which are shown in Table R1.

We have updated the information in Supplementary Table 1.

Table R1. CN and Al contents of the membranes.

Sample	CN nanosheet	ACN-1	ACN-2	ACN-3	ACN-4
Al ₃₀ /CN	0	0.29	0.80	1.00	0.62

5. the N₂ adsorption, the surface area and pore volume missed totally in this manuscript.

Response: Below we provide the N₂ sorption isotherms for our samples, along with the surface area and pore volume (Fig. R3). The surface area and pore volume increase with increasing polycation amount, pointing to the effective pillaring effect enabled by the polycations. A further increase of Al₃₀ results in the decrease of surface area and pore volume in ACN-4, this can be ascribed to the blocked pores (spaces) by Al₃₀, which tends to make the lamellar structure aggregated and less identifiable (as observed in the SEM images in Supplementary Fig. 9). We would also emphasize that the nitrogen sorption data can not be used to determine the interlayer channel for molecules or ions to transport in the widely reported 2D membrane family, instead, it only indicates the existence of available spaces for foreign species to access.

We have added the sorption data and plotted as new Supplementary Fig. 11, which was also described in “Characterizations” and “Supplementary Note 1”:

Page 3:

“11) Nitrogen sorption isotherms were recorded at 77 K using a Quantachrome Quadrasorb SI apparatus. Prior to the measurements, the samples were activated at 120 °C for 20 h under vacuum (0.5 Torr, 3 P Instruments Masterprep degassing device). The Brunauer-Emmett-Teller (BET) method was applied to calculate the specific surface area from adsorption branch data ($0.05 < P/P_0 < 0.35$). The pore volume was calculated from the amount of gas adsorbed at $P/P_0 = 0.995$.”

Page 4:

“The nitrogen sorption isotherms (Supplementary Fig. 11) verified that both surface area and pore volume increased with increasing Al_{30} amount, again pointing to the effective pillaring effect from the polycations. Further increasing Al_{30} resulted in the decrease of surface area and pore volume in ACN-4, this can be ascribed to the blocked pores by Al_{30} , which tended to make the lamellar structure aggregated and less identifiable (Supplementary Fig. 9i). It should be emphasized that the nitrogen sorption data cannot be used to determine the interlayer channel for molecules or ions to transport in 2D membranes, instead, it only indicates the existence of available spaces for foreign species to access.”

Fig. R3 Nitrogen sorption isotherms of CN and ACN samples recorded at 77 K. Inset shows the surface area and pore volume.

REVIEWERS' COMMENTS

Reviewer #1 (Remarks to the Author):

The authors have done a great effort in addressing all my concerns and the manuscript is now in a very good shape to be published in Nature Communication Journal.

Only a couple of suggestions/typos are left to be addressed:

- Fig. S29 should report the R-squared values of the fitting
- Page 5. "The E values of divalent ions (41.8 kJ mol⁻¹ for Ca²⁺ and 44.8 kJ mol⁻¹ for Mg²⁺) are much higher than that of monovalent ones (11.5,13.7 and 13.1 kJ mol⁻¹ for K⁺, Na⁺ and Li⁺, respectively)". Space is missing after "11.5,"

Reviewer #2 (Remarks to the Author):

I have reviewed the manuscript carefully and think the authors have addressed all my comments.

Reviewer #3 (Remarks to the Author):

The authors addressed all the questions of the reviewer. After modification, the manuscript has been well improved. I think that it probably merits publication in this journal.

Point-by-Point Response to Reviewers' Comments

Reviewer #1 (Remarks to the Author):

The authors have done a great effort in addressing all my concerns and the manuscript is now in a very good shape to be published in Nature Communication Journal.

Response: The authors appreciate the positive comments from Reviewer #1, the detailed responses are listed as follows.

1. Only a couple of suggestions/typos are left to be addressed:
- Fig. S29 should report the R-squared values of the fitting.

Response: The fitting with R^2 values were provided. Accordingly, the original Fig. S29 has been updated in Supplementary Information.

- Page 5. "The E values of divalent ions (41.8 kJ mol⁻¹ for Ca²⁺ and 44.8 kJ mol⁻¹ for Mg²⁺) are much higher than that of monovalent ones (11.5,13.7 and 13.1 kJ mol⁻¹ for K⁺, Na⁺ and Li⁺, respectively)". Space is missing after "11.5,"

Response: The spacing has been added after 11.5 in the main text.

Reviewer #2 (Remarks to the Author):

I have reviewed the manuscript carefully and think the authors have addressed all my comments.

Response: The authors appreciate the valuable comments from Reviewer #2, which helped improving the quality of this work.

Reviewer #3 (Remarks to the Author):

The authors addressed all the questions of the reviewer. After modification, the

manuscript has been well improved. I think that it probably merits publication in this journal.

Response: Once again, the authors would extend cordial thanks to Reviewer #3 for useful comments and suggestions.